# Determinants of Patient Delay in Diagnosis of Pulmonary Tuberculosis in Somali Pastoralist Setting of Ethiopia: A Matched Case-Control Study

**DOI:** 10.3390/ijerph16183391

**Published:** 2019-09-12

**Authors:** Fentabil Getnet, Meaza Demissie, Alemayehu Worku, Tesfaye Gobena, Berhanu Seyoum, Rea Tschopp, Christopher T. Andersen

**Affiliations:** 1Department of Public Health, Jigjiga University, Jigjiga 1020, Ethiopia; 2School of Public Health, Haramaya University, Harar 138, Ethiopia; 3Addis Continental Institute of Public Health, Addis Ababa 26751, Ethiopia; 4School of Public Health, Addis Ababa University, Addis Ababa 1176, Ethiopia; 5Armauer Hansen Research Institute, Addis Ababa 1005, Ethiopia; 6Swiss Tropical and Public Health Institute, 4002 Basel, Switzerland; 7University of Basel, 4051 Basel, Switzerland; 8Harvard TH Chan School of Public Health, Harvard University, Boston, MA 02138, USA

**Keywords:** patient delay, determinant, diagnosis, tuberculosis, pastoral, Ethiopia

## Abstract

***Background***: Healthcare-seeking behavior is the basis to ensure early diagnosis and treatment of tuberculosis (TB) in settings where most cases are diagnosed upon self-presentation to health facilities. Yet, many patients seek delayed healthcare. Thus, we aimed to identify the determinants of patient delay in diagnosis of pulmonary TB in Somali pastoralist area, Ethiopia. ***Methods***: A matched case-control study was conducted between December 2017 and October 2018. Cases were self-presented and newly diagnosed pulmonary TB patients aged ≥ 15 years who delayed > 30 days without healthcare provider consultation, and controls were patients with similar inclusion criteria but who consulted a healthcare provider within 30 days of illness; 216 cases sex-matched with 226 controls were interviewed using a pre-tested questionnaire. Hierarchical analysis was done using conditional logistic regression. ***Results***: After multilevel analysis, pastoralism, rural residence, poor knowledge of TB symptoms and expectation of self-healing were individual-related determinants. Mild-disease and manifesting a single symptom were disease-related, and >1 h walking distance to nearest facility and care-seeking from traditional/religious healers were health system-related determinants of patient delay > 30 days [*p* < 0.05]. ***Conclusion***: Expansion of TB services, mobile screening services, and arming community figures to identify and link presumptive cases can be effective strategies to improve case detection in pastoral settings.

## 1. Background

Tuberculosis (TB), caused by *Mycobacterium tuberculosis* (MTB), remains a global health problem. Worldwide, an estimated 10 million cases and 1.3 million deaths reported in 2017. In the same year, Ethiopia reported an estimated 172,000 new TB cases, which made the country 11th among the high burden and 4th in Africa [1]. The disease burden is higher in pastoralist communities of the country including the Somali Region [2]. The global End-TB strategy targets to end the epidemic by the upcoming decade using early detection of TB cases and prompt initiation of treatment as principal tools [3]. To achieve the global targets, the national TB control program (NTP) of Ethiopia implements passive case finding as the leading strategy. The term ‘passive’ implies the healthcare system identifies cases upon self-presentation of symptomatic patients to healthcare facilities [4].

This passive approach has not achieved the envisioned case detection rates in Ethiopia. Community-based studies revealed that the number of undetected TB cases in communities can be equal or beyond the number of notified cases [5,6]. The problem is more pronounced in Somali Regional State of Ethiopia where the case detection rate remained lowest (34%) in 2015 although the NTP targeted a 70% detection rate by 2015 [7] and still remains at 50% [8]. The slow progress in improving case detection could have resulted from diminished proportion of symptomatic cases coming to health facilities or poor performance of the health system in presuming and investigating notified cases [9]. In healthcare systems where passive case finding is predominant, the healthcare-seeking behavior of symptomatic cases is crucial to ensure timely identification of TB patients and initiation of treatment [10].

However, extreme delays of patients in seeking medical attention result in increased disease severity, prolonged patient suffering, adverse treatment outcomes, increased fatality and disease transmission [11,12]. Patients could delay from few days to months, some surpassing a year, until severe complications are manifested [13]. Reports highlighted that factors contributing to delayed healthcare seeking behavior are deep-rooted to individual, community and structural/health system facets but vary across different political, geographical, socio-cultural and epidemiological settings [14,15,16,17,18,19]. Identifying the determinants of patient delay in high-burden settings may help reducing these barriers, which delay presentation of presumptive TB cases to healthcare facilities.

Studies by Gele et al. highlighted armed conflict and limited access to TB services as the major barriers to healthcare seeking behavior in pastoral areas of Ethiopia [20,21]. The extent of patient delay remained unchanged despite a decreased extent of armed conflicts and substantial expansion of TB services. In such settings, the scattered settlement and mobility will leave a passive strategy as the leading case-detection approach [22]. This study, therefore, aimed to identify the existing determinants of patient delay in diagnosis of pulmonary TB in Somali Regional State of Ethiopia, predominantly inhabited by pastoralists.

## 2. Methods

### 2.1. Study Setting

The study was conducted in Kharamara, Dege-habour, Kebri-Daher and Gode hospitals, and Abilelie health center in Somali Regional State, Ethiopia. The facilities are the main TB care centers that serve majority of TB patients and cover wider geographical areas in the region. More than 85% of the total population lead nomadic or agro-pastoral way of life, and their livelihood relies mainly on livestock [23]. The regional healthcare system provides free TB diagnostic and treatment services based on the national TB guidelines [4].

### 2.2. Study Design and Population

A matched case-control study was conducted to identify the determinants of patient delay in seeking healthcare among patients with confirmed pulmonary TB (PTB). **Cases** were PTB patients aged ≥15 years who delayed more than 30 days without consulting a healthcare provider following the onset of pulmonary symptoms related to TB. Each case was matched individually with a control (1:1 ratio) using sex and from the same study facility. **Controls** were patients with similar inclusion criteria as cases except those who sought timely care within 30 days of the onset of pulmonary symptoms related to TB. We used 30 days’ cutoff to classify cases and controls based on a finding from China that depicted 30 days delay in care as the turning point at which the disease becomes critical and risk for transmission increases [24] although the NTP recommends people with two or more weeks of cough to seek TB care [4]. Self-presented and newly diagnosed patients aged ≥15 years between December 2017 and October 2018 were included regardless of their smear status and treatment history. Screening questions were used to identify eligible patients. Patients identified using active case finding efforts were excluded, which include human immunodeficiency virus (HIV) co-infected patients identified during follow up screening, and those identified by health extension workers (HEWs) during home visits.

### 2.3. Sample Size and Sampling Technique

The minimum sample size calculated was 430 (215 cases and 215 controls) using OpenEpi303 software for case-control studies. Assumptions considered include 95% confidence interval (CI), power of 90%, case to control ratio of 1:1, and 19.0% of controls and 33.9% of cases sought a first consultation from a religious/traditional provider in Ethiopia [25], 5% precision and 10% non-response rate. Upcoming patients were recruited prospectively and classified into cases and controls based on their delay (starting from the first to the end date of data collection). The final sample size was 216 cases and 226 controls (total = 442). To minimize misclassification of cases and controls due to recall bias, we recruited the patients as they arrived at a directly observed treatment-short course (DOTs) unit while expected to have fresh memory.

### 2.4. Data Collection

Data on socio-demographic, access to services, care seeking activities, knowledge, perceived barriers to care (*myths/perceptions*) and clinical characteristics were collected through patient interview using pre-tested and structured questionnaire. The questions were adapted from similar studies in Eastern Mediterranean WHO region [26] and Ethiopia [20,25,27]. In addition, data on acid-fast bacilli (AFB) and X-ray results, HIV status, and co-morbidities were collected from patient records. Nurses in the respective DOTs clinics conducted interviews and record reviews using the questionnaire that was translated into local Somali and Amharic languages by language experts. To minimize misclassification due to recall bias, patients were asked the first dates of illness onset, provider consultation, and diagnosis (instead of asking them to remember the number of days they stayed without provider consultation). If uncertain, patients were encouraged to link dates with critical events (religious or cultural). The durations of delay were calculated afterwards (time span from onset of symptoms to first consultation). Cough or another main symptom that compelled the patients to seek care (*in the absence of cough*) was used as a benchmark of symptom onset. Information from participants was also crosschecked using record review. All interviews were made prior to health education and treatment initiation, which could influence the awareness level of patients.

### 2.5. Measurement of Tuberculosis (TB)-Related Knowledge

The knowledge section of the questionnaire had 20 questions grouped into three categories: comprehensive TB knowledge (*seven questions on cause, transmission and treatment*), knowledge of main TB symptoms (*nine questions*), and knowledge of respiratory TB symptoms (*four questions*). The questions were intended to assess the patients’ knowledge of TB plausibly influencing their healthcare seeking behavior. *Yes/correct* responses were labeled as “1”, and *incorrect/no/I don’t know* responses were labeled as “0”. The scores were added up to create knowledge ranking for the aforementioned categories. The pooled scores of questions under each knowledge category were classified into poor and satisfactory knowledge using mean score values.

### 2.6. Measurement of Perceived Barriers

Perceived barriers were assessed using 10 questions, five questions on common myths/misconceptions towards the disease and its remedy, and the rest on patients’ perception towards their illness that are supposedly hindering patients’ healthcare seeking behavior. The questions had negative dimension with dichotomous responses of “*Yes, I believe*” and “*No, I don’t believe’* that denote negative and positive perceptions, respectively.

### 2.7. Data Processing and Analysis

All components of data were double entered and validated using EpiData 3.1 and analyzed using Stata version 14. Data were summarized in frequencies/proportions and mean/median to describe the explanatory variables of cases and controls. Odds ratios with 95% confidence intervals (CI) were used to compare the odds of delayed care seeking (patient delay >30 days or ***case***) between categories of plausible determinants, given timely healthcare seeking (patient delay ≤30 days) as reference (control). Hierarchical/multilevel analysis was done using conditional logistic regression models for matched data. The variables in the study were categorized into three blocks of factors: socio-demographic, clinical, and behavioral (*knowledge, perceived barriers and initial care practice*). We ran three separate models (model 1, 2 and 3) for each block using variables with *p*-value ≤ 0.2 in bivariate analysis. At last, the final model (model 4) was fitted using the variables with *p* ≤0.05 in each model.

### 2.8. Operational/Standard Definitions of Terms

Pulmonary Tuberculosis (PTB): is a patient with lung TB and confirmed by microbiological (AFB/culture/GeneXpert) and clinical diagnosis. A smear positive patient is confirmed if at least one AFB positive smear; A smear negative patient is diagnosed if: at least two AFB smear negative results, no response to a course of broad-spectrum antibiotics, again two AFB negative smears and radiological abnormalities consistent with TB; or two AFB negative smears but culture or GeneXpert positive [4].

Patient delay: is defined as the period (in days) from onset of pulmonary symptom(s) particularly cough or another main symptom related to PTB (*in the absence of cough*) until the date the patient first consults a healthcare provider.

Healthcare provider: refers to a nurse, clinical officer or doctor at health center and hospital, or private facilities with TB services. 

Newly diagnosed patient: is a patient who was diagnosed with pulmonary TB prospectively during the study period. This excludes patients who were on treatment.

Self-presented patient: is a PTB patient who sought initial care for his/her pulmonary illness without being referred by healthcare providers (excluding patients identified by active case finding).

New case: is a patient who has never had treatment for TB before or has not yet initiated anti-TB treatment.

Retreatment patient: is a patient who was treated for any form of TB before but has developed the disease again (relapse, default or therapy failure of the first regimen).

## 3. Results

### 3.1. Socio-Demographic and Clinical Characteristics

Overall, 442 PTB patients (216 cases and 226 controls) aged ≥ 15 years were included in the study. The age of participants ranged 15 to 82 years with median of 30 (inter-quartile range (IQR) 23–50) years. The majority was male (62.7%) and illiterate (62.0%). Nearly half (46.6%) rely on pastoral livelihood, of which 35.9% were nomadic. Half (50%) of the patients walked over 40 minutes and 25% walked two or more hours to reach any nearest health facility with/without TB diagnostic services. Four pastoralist patients travelled for almost one full day to access TB services. The majority was new cases (90.0%) and smear negative (57.7%); 3.4% were co-infected other pulmonary diseases mainly TB-pneumonia, and 2.3% were HIV positive (Table 1).

### 3.2. Knowledge and Misconceptions Towards TB

Of the total study participants, the majority had known TB as a disease before they got ill (84.6%), and had a conception that TB is curable using modern medicine (81.0%), fatal (75.1%) and transmissible disease (59.1%). Nevertheless, 84.4% did not know about the Bacillus Calmette-Guerin (BCG) vaccine. The means of TB transmission mentioned by patients who claimed TB is transmissible were via airborne droplets (84.6%), contaminated food or drink (6.6%), sexual intercourse (5.8%), and others (3.0%). Pertaining to knowledge of TB symptoms, patients had a conception of cough more than 2 weeks (86.4%), chest pain (62.7%), difficulty while breathing (33.7%), blood in sputum (18.1%), excessive sweating during night (43.0%), tiredness (53.2%), loss of body weight (39.4%) and fever (36.2%) as symptoms that patients with pulmonary TB exhibit. Regarding misconceptions related to TB, patients had a perception that TB is caused by exposure to cold air (62.2%), acquired due to evil curse/bad luck (24.2%), can be cured using traditional medicine (10.9%), and people with TB are always HIV positive (18.8%).

### 3.3. Healthcare Seeking Behavior

The median delay of cases (patient delay >30 days) was 50 days (IQR: 40–72), ranged 31 to 330 days, and of controls (patient delay ≤30 days) was 20 days (IQR: 14–25), ranged 4 to 30 days. As initial points of care, 76.5% patients presented to health centers or hospitals, 12.9% attempted self-medication/traditional remedy, 10.6% sought care from low-level (*drugs vendors/HEWs*), and 13.8% visited traditional/religious healers as initial point of care or any time in the course of the disease (Table 2). Patients who did not seek initial care from healthcare providers mentioned various reasons for opting for informal care such as long waiting time/crowdedness (32.7%), distance (25%), prior bad experience in public health facilities (24.0%), lack of confidence in the quality of services (9.6%), and fear of service costs (5.8%) in health facilities. Overall, participants contacted a mean of 2.3 ± 1.2 healthcare providers before their disease was diagnosed as TB.

### 3.4. Determinants of Patient Delay in Diagnosis

The following were found significant among the socio-demographic, behavioral (*knowledge, perceived barriers and practice*), and clinical variables included in the final model. The odds of patient delay >30 days was higher in pastoral [adjusted odds ratio (AOR) (95%CI) = 2.1(1.2–3.6)] and rural [AOR (95%CI) = 2.1 (1.3–3.7)] patients, and those who walked over an hour to access any health facility [AOR (95%CI) = 3.2(1.9–5.6)] compared to their respective counterparts. Patients who had poor knowledge of TB symptoms [AOR (95%CI) = 2.7(1.5–4.8)], thought symptoms would go away gradually [AOR (95%CI) = 2.1(1.3–3.5)], and visited traditional/religious healers [AOR (95%CI) = 2.1(1.1–4.2)] were more likely to delay longer than 30 days without provider consultation. Moreover, patients with mild severity of disease [AOR (95%CI) = 1.6 (1.01–2.6)] and had single respiratory symptom during onset of disease [AOR (95%CI) = 2.2 (1.3–3.9)] were also more likely to delay longer than 30 days, but patients with multiple general or respiratory symptoms at onset were less likely to delay (Table 2).

## 4. Discussion

The findings revealed that patient delay was attributed to a set of determinants pertaining to the individual patient, disease condition or health system characteristics. Individual factors influenced the patients’ intent to seek healthcare or limited their access to health facilities/services. These include pastoral lifestyle, rural residence, poor knowledge, and considering symptoms as less severe. Disease-related factors were severity and multiplicity of symptoms that patients manifested upon disease onset; whereas, the health system elements comprise long distance to access health facilities and visits to traditional healers.

Our study showed that rural and/or pastoral patients were more likely to delay more than a month without medical attention. The reason may relate to long distance to or absence of frontline diagnostic services in rural settings. About 60% of rural/pastoral patients traveled longer than an hour to reach any kind of health facility, and some patients traveled a full day to reach them. Rural/pastoral subjects had access to health posts and health centers rather than secondary/tertiary care centers in cities, but the health posts (*first level in the health system*) are not intended to provide TB diagnostic services [28] and about two-thirds of health centers did not provide TB diagnostic and treatment services in the area [8]. Some also closed operations in dry seasons due to transhumance patterns of pastoralists [21]. This implies that lack of diagnostic services in firsthand points of care can influence the care-seeking behavior, as patients may not be interested in visiting health facilities with limited services. In such situations, patients may instead opt for informal cure [16] and are forced to several other visits searching for appropriate facility (*up to six visits in this study*) that leads to extra delay [18,29]. Moreover, limited accessibility of services can also affect the implementation of DOTs that demands daily presentation of patients in health facilities [28].

Along with challenges to accessing TB services, poor knowledge about symptoms and belief that symptoms would go away gradually were associated with higher odds of patient delay. This shows that patients with poor awareness of the disease and its implication suffer longer periods without seeking medical attention. This may be because patients with poor knowledge of symptoms associate the symptoms they have with common respiratory syndromes that are not regarded as serious, and such patients believe TB is a serious disease with special symptoms [30]. Our data also reveal that patients with single symptom and mild severity of disease were more likely to delay longer than a month. Hence, patients with poor knowledge and uncomplicated symptoms may hope for gradual healing or look for alternative medicines, self-medication or herbal remedies, but do not seek medical attention unless severe complications are manifested [19,31].

More to the point, traditional/religious healers served as initial points of TB care for 13.8% patients and substantially contributed to excessive patient delays. Long waiting time, long distance, bad experience, and lack of confidence in the quality of health services were mentioned as reasons for visiting informal healers and drug vendors as initial points of care. Similar studies also supported the idea that patients with poor knowledge about TB’s cause or associating TB with curse/bad luck have confidence on and prefer traditional/religious healers [13,32]. This implies that traditional/religious healers will remain as preferred points of care for deprived patients and communities. Therefore, involving traditional/religious healers in TB services can help to identify presumptive cases early and immediate linkage to TB care units, to avoid inappropriate therapy, and to tackle poor knowledge and misconceptions in communities. Patient delay can then be reduced, and case detection can be improved ultimately [10,33].

Nonetheless, other studies found patient delay to be associated with the following factors; fear of social discrimination, low-income, illiteracy, sex (*varies among studies*), misconception about PTB cause, initial visit to private practitioners/drug vendors, previous history of TB, unemployment, urban residence and large family size, to mention some among many others [25,34,35,36]. The inconsistencies may arise from socio-cultural, epidemiological and health system-based differences across settings and countries as well as methodological variations among studies.

This study might be subjected to recall bias since we relied on patients’ reports of the onset of symptoms and dates they sought care that works for both the cases and controls. A day’s difference from cutoff (i.e 30 or 31) might misclassify cases as controls, or vice versa, although it is random.

## 5. Conclusions

Several individual, clinical and health system-related factors appeared to influence healthcare-seeking behavior of pulmonary TB patients in Somali pastoralist setting, Eastern Ethiopia. A long distance to health facilities, rural residence, pastoral livelihood, poor knowledge of TB symptoms, interpreting symptoms as less severe, and care seeking from traditional/religious healers are independent determinants of patient delay in diagnosis of PTB in our study. Limited access or lack of TB services in primary healthcare units remains problematic in pastoral/rural communities. Therefore, expansion of TB services to primary healthcare units, mobile/outreach screening services to remote pastoral villages, and promotional activities should be central to the efforts aimed at improving early case detection in pastoralist setups of the country. Moreover, many pastoralist villages have prominent clan and/or religious leaders so that involving them to identify presumptive cases and link to health facilities may be an effective strategy to improve timely presentation of symptomatic cases in pastoral communities.

## Figures and Tables

**Table 1 ijerph-16-03391-t001:** Clinical characteristics of pulmonary tuberculosis (TB) patients in Somali Region, Ethiopia, December 2017 to October 2018 (*controls* = 226, *cases* = 216).

Characteristics	Frequency (%)		*p*-Value
	Controls (*n* = 226)	Cases (*n* = 216)	
Sex			0.95
Female	84 (37.2)	81 (37.5)	
Male	142 (62.8)	135 (62.5)	
Age group (years)			0.74
15–23	60 (26.5)	57 (26.4)	
24–30	58 (25.7)	57 (26.4)	
31–50	60 (26.5)	64 (29.6)	
51+	48 (21.3)	38 (17.6)	
Literacy level			0.67
Illiterate	134 (59.3)	140 (64.8)	
Primary	26 (11.5)	20 (9.3)	
Secondary	34 (15.0)	30 (13.9)	
Tertiary	32 (14.2)	26 (12.0)	
Marital status			0.92
Single	67 (29.7)	65 (30.1)	
Married	141 (62.4)	130 (60.2)	
Divorced/separated	8 (3.5)	9 (4.2)	
Widowed	10 (4.4)	12 (5.5)	
Residence			0.04
Rural	102 (45.1)	117 (54.2)	
Urban	123 (54.4)	96 (44.4)	
Refugee/displaced ᶲᶲ	1 (0.5) ᶲᶲ	3 (1.4) ᶲᶲ	
Livelihood			0.03
Pastoralism	94 (41.6)	112 (51.9)	
Non-pastoralism	132 (58.4)	104 (48.1)	
Income			0.07
Saving	29 (12.8)	28 (13.0)	
Income = expense	148 (65.5)	159 (73.6)	
Indebt	49 (21.7)	29 (13.4)	
One-way walking time to nearest HF			<0.001
Within half an hour	125 (55.3)	89 (41.2)	
Half to an hour	53 (23.5)	38 (17.6)	
One to two hours	24 (10.6)	36 (16.7)	
More than two hours	24 (10.6)	53 (24.5)	
Physical conditionat 1st consultation			0.83
Good	34 (15.0)	29 (13.4)	
Ambulatory	185 (81.9)	180 (83.3)	
Bedridden	7 (3.1)	7 (3.3)	
Symptoms at disease onset			
Cough	217 (96.0)	197 (91.2)	0.04
Haemoptysis	19 (8.4)	0	<0.001
Chest pain	134 (59.3)	119 (55.1)	0.37
Breathing difficulty	25 (11.1)	28 (13.0)	0.53
Weight loss	81 (35.8)	76 (35.2)	0.88
Night Sweating	76 (33.6)	67 (31.0)	0.56
Fatigue	86 (38.1)	107 (49.5)	0.02
Fever	58 (25.7)	23 (10.7)	<0.001
Loss of appetite	49 (21.7)	53 (24.5)	0.47
TB category			0.63
New	205 (90.7)	193 (89.4)	
Retreatment	21 (9.3)	23 (10.6)	
Smear status			<0.001
Positive	76 (3.6)	111 (51.4)	
Negative	150 (66.4)	105 (48.6)	
HIV status			0.27
Positive	4 (1.8)	6 (2.8)	
Negative	222 (98.2)	208 (96.3)	
Unknown	0	2 (0.9)	
Pulmonary co-morbidity			0.27
None	221 (97.8)	206 (95.4)	
Pneumonia	3 (1.3)	7 (3.2)	
Asthma	1 (0.4	3 (1.4)	
Bronchitis	1 (0.4)	0	
Chronic disease *(HTN/HD/RD)*			0.36
Yes	15 (6.6)	10 (4.6)	
None	211 (93.4)	206 (95.4)	
Diabetes mellitus			0.68
Yes	9 (4.0)	7 (3.2)	
No	217 (96.0)	209 (96.8)	

Key: ᶲᶲ indicates the response category was not included in the comparison. The percentage for each symptom at onset is out of the total cases and controls separately. Abbreviation: HTN (Hypertension); HD (Heart Disease); RD (Rental Disease).

**Table 2 ijerph-16-03391-t002:** Patient delay >30 days among pulmonary TB patients in Somali Region, Ethiopia, December 2017 to October 2018, *Hierarchical Analysis* (*control* = 226, *cases* = 216).

Characteristics	Controls (≤30 days)	Case (>30 days)	Model 1	Model 2	Model 3	Final Model
	*n* (%)	*n* (%)	AOR (95%CI)	AOR (95%CI)	AOR (95%CI)	AOR (95%CI)
**Socio-demographic factors (Block 1)**						
Educational level						
Illiterate	134 (59.3)	140 (64.8)	1.4 (0.8,2.5)			
Literate (formal)	92 (40.7)	76 (35.2)	1			
Residence						
Rural	102 (45.1)	117 (54.2)	1.8 (1.04,3.2)			2.1 (1.3, 3.7)
Urban	123 (54.4)	96 (44.4)	1			1
Refugee/displaced ᶲᶲ	1 (0.5) ᶲᶲ	3 (1.4) ᶲᶲ	ᶲᶲ			ᶲᶲ
Livelihood						
Pastoralism	94 (41.6)	112 (51.9)	1.9 (1.1,3.5)			2.1 (1.2, 3.6)
Non-pastoralism	132 (58.4)	104 (48.1)	1			1
Income						
Indebt	49 (21.7)	29 (13.4)	0.5 (0.2,1.1)			
Income = expense	148 (65.5)	159 (73.6)	1.1 (0.5,2.0)			
Saving	29 (12.8)	28 (13.0)	1			
Family size (average)						
>6 members	91 (40.3)	102 (47.2)	1.3 (0.9,2.0)			
≤6 members	135 (59.7)	114 (52.8)	1			
Smoking (all male, *n* = 277)						
Ever smoker	17 (12.0)	28 (20.7)	1.8 (0.9,3.7)			
Never smoker	125 (88.0)	107 (79.3)	1			
Khat chewing						
Ever chewer	25 (11.1)	33 (15.3)	1.7 (0.9,3.3)			
Never chewer	201 (88.9)	183 (84.7)	1			
Walking distance to nearest HF						
More than an hour	48 (21.2)	89 (41.2)	3.2 (1.8,5.6)			3.2 (1.9,5.6)
Within an hour	178 (78.8)	127 (58.8)	1			1
**Knowledge, perceived barriers and Initial Care seeking behavior (Block 2)**						
Knowledge of main TB symptoms						
Poor	54 (23.9)	73 (33.8)		2.5 (1.4, 4.3)		2.7 (1.5, 4.8)
Satisfactory	172 (76.1)	143 (66.2)		1		1
TB is owing to evil/bad luck						
Yes	48 (21.2)	59 (27.3)		1.4 (0.7, 2.6)		
No	178 (78.8)	157 (72.7)		1		
Cold air exposure as TB cause						
Yes	131 (58.0)	144 (66.7)		1.5 (0.9, 2.5)		
No	95 (42.0)	72 (33.3)		1		
Hoped illness go- away gradually						
Yes	103 (45.6)	128 (59.3)		2.0 (1.2, 3.3)		2.1 (1.3, 3.5)
No	123 (54.4)	88 (40.7)		1		1
First action upon recognized illness						
Visit HC/Hospital	180 (79.6)	158 (73.2)		1		
Visit low-level HF	21 (9.3)	26 (12.0)		1.2 (0.6, 2.5)		
Self/ traditional remedy	25 (11.1)	32 (14.8)		1.03 (0.5, 2.2)		
Visit traditional/religious healer						
Yes	23 (10.2)	38 (17.6)		2.1 (1.1, 4.1)		2.1 (1.1, 4.2)
No	203 (89.8)	178 (82.4)		1		1
**Pulmonary manifestations and co-morbidities (Block 3)**						
Presence of Cough						
Yes	217 (96.0)	196 (90.7)			0.5 (0.2,1.2)	
No	9 (4.0)	20 (9.3)			1	
Symptoms at disease onset *(Of 8)*						
Single or two	72 (31.9)	89 (41.2)			1.6 (1.01,2.5)	1.8 (1.1,2.9)
Three or more	154 (68.1)	127 (58.8)			1	1
Multiplicity of respiratory symptoms *	1.7 ± 0.7 *	1.6 ± 0.7 *			0.6 (0.4,0.9)	0.5 (0.3,0.7)
Respiratory symptoms *(Of 4)*						
Single	87 (38.5)	100 (46.3)			1.7 (1.03,2.8)	2.2 (1.3,3.9)
Two or more	139 (61.5)	116 (53.7)			1	1
Severity of Disease						
Mild	120 (53.1)	136 (63.0)			1.6 (1.04,2.5)	1.6 (1.01,2.6)
Moderate/severe	106 (46.9)	80 (37.0)			1	1
Pulmonary co-morbidity						
Yes	5 (2.2)	10 (4.6)			2.0 (0.6,6.4)	
No	221 (97.8)	206 (95.4)			1	

Key: ᶲᶲ (The response category is dropped out of analysis); * (The explanatory variable is numeric: the values are mean ± standard deviation); -- (The variable is not included in the final model); Low level HF: includes drug vendor, health post or lower private clinics. Abbreviations: HCP (HealthCare provider); COR (Crude Odds Ratio); AOR (Adjusted Odds ratio); n (Number); HC (Health Center); HF (Health facility).

## Data Availability

The data supporting the conclusions of this article is included within the article. The collected data contain confidential information, and consent has not been obtained for public sharing of raw data with identifiers. However, the datasets used and/or analyzed are available at the hands of the corresponding author and can be shared upon reasonable requests.

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
