# Peer review of "Determinants of Patient Delay in Diagnosis of Pulmonary Tuberculosis in Somali Pastoralist Setting of Ethiopia: A Matched Case-Control Study"

_ijerph, 2019, doi:10.3390/ijerph16183391_

Round 1

Reviewer 1 Report

This study has given some information, however, I have some questions as followings:  

How do you define onset of illness in this study? You selected patients who sought care within 30 days as control and patients who sought care more than 30 days as case. You used 30 days cutoff to classify case and control due to 30 days delay in care is the turning point of disease. My question is do you think their characteristics are different among patients who delay 10 days, 20 days, 30 days, 60 days, and more? Their risk factors for delay seeking care are different?   What is the strength of this study?   

Reviewer 2 Report

This is a study where the determinants of patients delay for early diagnosis of pulmonary tb in a rural setting of Ethiopia, including nomads is investigated. The use of determinant for time is 30 days, as detected to be an important timeline for progress of the disease in previous studies in other parts of the globe. The national programme is built on self seeking and this study shows that this is not working well and suggests solutions such as mobile screening services in rural settings of the country. The study is of importance as it can influence a national programme of a disease with high mortality and morbidity and it concludes the proper scientific methodology used with suggestions for improvements including involvement of local healers.

My only concern is that there is no genderdimension in the analysis of the data. There is a predominance of men among the self-seekers and it is not discussed, nor is the different determinants, well presented in the table presented gender divided and discussed. Tb is a disease where gender differences have long been published and discussed from different perspectives. I suggest a paragrpah is added in the discussion in this paper and that the data the authors must have are presented also gender divided in another manuscript; there is plenty of more interesting data here to be analysed, discussed and published.

Reviewer 3 Report

Thanks for your interesting study, which is highly relevant. 

Some general points: much of the background can be shortened, some of it is redundant. Ordering sentences and paragraphs can make your article more readable. I suggest using more scientific wording and consistent terminology. Past tense should be used throughout. In the conclusion you can specify some of the measures you would like to implenent as a consequence of your data

Some suggestions:

Line/suggestion

20/whats the outcome you want to prevent - general aim?

23/without-healt care- provider consultation

24/what are "the same patients"

27/knowledge: define: TB-related knowledge as opposed to genereal education

27/use "disease severity" as opposted to "mild"

31/wise: unscientific wording!

27/2017: use updated literature

39/communitIES

41/change to "treatment as principle tools"

47/worse --> more pronounced

48/target - whose?

48 "has till remained" --> remains

49/ progress in IMPROVING case

50/suspected --> presuming

51/takes the lead-->is predominant

52/plays the utmost role-->is crucial for

53/presentation of symtomatic cases --> identification of TB cases

54/remove "of TB patients"

55/"have been .... care worsens" --> result in increased

56/prolongs-->prolonged

57/ change to "adverse treatment outcomes, increased fatality and disease transmission, especially in patoral areas"

62/change to "may help reducing these, which delay presentation of presumptive TB cases" PARAGRAPH

64/area (which?)

65/ change to "remained unchanged despite a decreased extent of .... and despite substantial...", remove "since then"

66/remove "moreover .... areas"

67/ after settings add (scattered settlements, low mobility)

68/ after "leading" add "case detection approach", remove "and scattered ... efforts"

75/areaS

77/ remove "The nomadic ... mobility"

Section 2.2: change order: 1 Case definition, 2 matching procedure and definition of controls

82/ "A" matched case ....

86/"same patients" NO! not the same... "were patients with similar inclusion criteria as cases, except ..."

88/"gets worse" unscientific! 

90/self-presented: were these all TB-diagnosed?

100/ add () from "starting" to "collection"

101/"became"-->"was"

Section 2.4: 

smear: better use AFB/specific staining method

113/remove "critical"; insert () from "instead of" to "consultation"

114/ remove "when patients were", change to "if uncertain,"

115/ remove "of the dates, they", change to "patients"; add "dates" after "link"; remove "to improve... measurement"

116/ remove "using", change "critical ..." to "(time span from onset of symptoms to first consultation)

121/ specify what knowledge, e.g. "TB-related knowledge"

127/"together" --> "up"

Section 2.8:

clarify! A) microbiological (smear OR culture OR Xpert) B) clinical diagnosis (none of A) )

155/ after "provider": PARAGRAPH

160/ remove "told (" and "to do so"; then remove "This" and put the rest into (): "(excluding patients identified by active case finding)", remove the remainder of the sentence

165/ change to: "again (relapse, default or therapy failure of the first regimen)

169/ rangeD

Section 3.1: ORDER: 1. epi data on patients, 2. TB-related data, 3. co-infection

170/ specify according to case definition: how many were confirmed cases, i.e. smear negative, but culture positive

172/ pneumonia: acute? TB-pneumonia? no TB? 

Section 3.2: better use terms like "patents had a conception of" rather than "responded, said, believed"

183/manifest --> exhibit

194/remove "faraway"; experience (with WHAT??)

211/remove "the" before "individual" and "are"

212/remove "characteristics of the patient"; "limit-->"limited"

214/change to "disease-related factors were severity"

215/change to manifested upon disease onset

217/218 I don't understand what you are trying to say

219/your opinion? always clarify: conclusion drawn from data (add specific figures) or conclusion merely author opinion?/assumption?

220/221 remove „at which … happen“

222/change to „subjects had access to health posts rather than secondary/tertiary care centers in cities“

223/ do -> did

224-226/ change to „Health posts (first…) are not intended to …“ (shorten!)

234/menace —> implications

235/ is -> may be, later remove may

242/remove inevitable, include a specific figure

245/evidences —> singular! And what evidence??

247/use present tense

251/ halted —> reduced; intensified -> improved

252/ nonetheless, other studies found patient delay to be assc. With the following factors; no list factors…

265/ past tense

266/ add „in our study“ after „PTB“

269/ don’t make generalizations, rather say „many pastoralist villages“

270/equip —> involving 

271/ wise…. Scientists aren’t wise ;) rather say „facilities may be an effective strategy“

Round 2

Reviewer 1 Report

This manuscript has been revised by authors according to the suggestions of reviewers. This version has much improvement compared to  the previous version.